# Microwave-Assisted Synthesis, Structural Characterization and Assessment of the Antibacterial Activity of Some New Aminopyridine, Pyrrolidine, Piperidine and Morpholine Acetamides

**DOI:** 10.3390/molecules26030533

**Published:** 2021-01-20

**Authors:** Abdulmajeed S. H. Alsamarrai, Saba S. Abdulghani

**Affiliations:** Department of Chemistry, College of Applied Sciences, University of Samarra, Salahaldin 13/1333, Iraq; sabasabdulghani@gmail.com

**Keywords:** acetamide, pyridine, pyrrolidine, piperidine, morpholine, antibacterial activity, heterocycles

## Abstract

A series of new acetamide derivatives **22**–**28** of primary and secondary amines and *para*-toluene sulphinate sodium salt have been synthesized under microwave irradiation and assessed in vitro for their antibacterial activity against one Gram-positive and two Gram-negative bacterial species such as *S. pyogenes*, *E. coli*, and *P. mirabilis* using the Mueller-Hinton Agar diffusion (well diffusion) method. The synthesized compounds with significant differences in inhibition diameters and MICs were compared with those of amoxicillin, ampicillin, cephalothin, azithromycin and doxycycline. All of the evaluated acetamide derivatives were used with varying inhibition concentrations of 6.25, 12.5, 37.5, 62.5, 87.5, 112.5 and 125 µg/mL. The results show that the most important antibacterial properties were displayed by the synthetic compounds **22** and **24**, both of bear a *para*-chlorophenyl moiety incorporated into the 2-position moiety of acetamide **1**. The molecular structures of the new compounds were determined using the FT-IR and ^1^H-NMR techniques.

## 1. Introduction

Several acetamide derivatives have been reported to act as antimicrobial agents which have been shown to be highly effective against Gram-positive and Gram-negative species [1,2,3,4,5]. There are many reports on the use of α-chloroacetamides as useful building blocks for the synthesis of complex heterocyclic compounds. Further possible biological effects have also been shown by derivatives of the acetamide moiety **1** (Figure 1), notably as antimalarial [6], anticancer [7], anti-diabetic [8], anti-tuberculosis [9] and anti-inflammatory agents [10], as well as in industrial applications such as stabilizer synthesis [11], plastic releasing agents [12], film [13], surfactants and soldering flux [14], organic fibers [15] and dyes [16].

The compounds atorvastatin (**2**) [17], lidocaine (**3**) [18], paracetamol (**4**) [19], amoxicillin (**5**) [20], levobupivacaine (**6**) [21] are the most widely used drugs in medicine that contain the acetamide moiety **1** (Figure 1). Pyrrolidine, piperidine, morpholine, and piperazine are secondary amines that are very valuable saturated heterocyclic compounds because of their wide and diverse range of promising biological activities, such as antibacterial and anti-inflammatory activity [22,23,24,25,26]. In addition, many derivatives of **1** containing secondary cyclic amines linked to the 1 or 2-positions were reported to be anticonvulsant agents. For instance, anticonvulsant compounds such as **7** and **8** are used to treat epilepsy [27], while the combination of compounds **9** and **10** is used to inhibit kinase enzyme and antihistamines [28].

Pyridine heterocycles are present in certain biologically active molecules (Figure 2). One of the pyridine-containing medications is Lunesta with the active ingredient **11** that is used to treat insomnia, and pioglitazone **12** which is effective in diabetes care.

In addition, previous studies conducted by Nakamoto et al. [29] and Tang et al. [30] evaluated the compounds **13** and **14** as an antifungal and antituberculous agent, respectively.

The synthesis of compounds with broad antibacterial activity is required because of the extensive use of antibiotics in medicine and the prevailing resistance of microorganisms. Conventional organic preparation methods are too sluggish to meet the need for the rapid synthesis of these compounds. Nowadays microwave irradiation represents an interesting technique to be applied in organic synthesis [31]. Herein, we report the use of microwaves in the synthesis of seven new acetamide derivatives **22**–**28** by reaction of chloroacetyl chloride **15** with primary, secondary amines and *para*-toluene sulphinate sodium salts. In addition, the aim of this study was to assess the susceptibility profile of bacterial isolates to the synthesized compounds **22**–**28** and reference antibiotics, and to determine their minimum inhibition concentrations (MICs). The structures of compounds **22**–**28** were determined by ^1^H-NMR, and FT-IR and elemental analysis. Table 1 shows the approach followed.

## 2. Results and Discussion

### 2.1. Chemistry

We are engaged in intensive efforts to synthesize nitrogen-heterocyclic compounds in high yields. Following a previously described study [32], the α-chloroacetamide intermediates **16**–**21** and the new targeted derivatives **22**–**28** were designed and synthesized by introducing *para*-toluene sulphinate, pyrrolidinyl, morpholinyl, and piperidinyl moieties into the acetamide scaffold as can be seen in Scheme 1.

Over the past thirty years, microwave irradiation has been used to enhance reaction rates [33,34]. We have thus exploited the advance of microwave technology to accelerate the creation of C-N bonds. For the synthesis of the precursors **16**–**21**, the process was performed under mild conditions involving the addition of chloroacetyl chloride **15** to a mixture of amines and *para*-toluene sulphinate sodium salt in dry CH_2_Cl_2_ at 0 °C and the isolated yields of these compounds ranged from 60 to 90% (see Table 1). The synthesis of targeted compounds **22**–**28** was achieved through two routes: conventional heating and microwave irradiation. The heating process involved precursors **16**–**21**, amine, and sodium *para*-toluene sulphinate reactions in a CH_3_CN solution containing Et_3_N as a catalyst giving the compounds **22**–**28** in moderate yields reaching 60% at 70 °C (see Scheme 1 and Table 1), On the other hand, treatment of the precursors **16**–**21** with amines as well as sodium *para*-toluene sulphinate in dry CH_3_CN provided the desired compounds **22**–**28** at 65–70 °C in good yields under microwave irradiation. The reactions took 5–10 min to complete and one of the advantages of this technique is allowed us to isolate clean products in good yields and without side products (see Table 1 and Table 2 for the corresponding ^1^H-NMR data).

### 2.2. In Vitro Antibacterial Activity Testing

Instructive bacterial infections such as tuberculosis, urinary tract infection, pneumonia, brain abscess, phyaryngitis, and tonsillitis are a series of life-threatening diseases commonly recognized in immunocompromised patients. Such infectious diseases are becoming more and more resistant to antibiotics. Resistance to microbial drugs is an inevitable consequence of the overuse of antibacterial drugs. Susceptibility to antibiotics by pathogens has now become a crucial factor in the effectiveness of chosen antibacterial drugs. Amoxicillin, ampicillin, doxycycline, azithromycin and cephalothin, the five reference antibiotics used in this research, are commonly used for the treatment of infections. Researchers continue to seek new antimicrobial agents. This research was carried out to determine the susceptibility of certain bacterial organisms toward seven newly synthesized compounds. The synthesized compounds **22**–**28** have been evaluated for antibacterial activity in comparison with the abovementioned reference antibiotics.

Acetamide derivatives are well known to have a broad range of antibacterial actions [35]. With regards to our recent research [32] on the synthesis of acetamide derivatives and the assessment of their biological role, in this in vitro study, three species of microorganisms were included, and all isolates of the three species were susceptible to the synthesized compounds **22**–**28** at MICs greater than 6.25 μg/mL.

Tested compounds **22** and **23**, exhibited inhibition zones ranging from 6.0 to 8.4 mm at MIC of 12.5 μg/mL against the Gram-negative species *E. coli* and *P. mirabilis* that cause urinary tract infections (UTI) in patients, while isolates of these bacterial species were not susceptible to compounds **26** and **28** even at an MIC of 37.5 μg/mL. Compounds **24**, **25**, and **27** showed inhibition zones ranging from 6 to 14 mm at a MIC of 37.5 μg/mL (Figure 3 and Figure 4, and Appendix A). These intermediate results are in agreement with other authors who have reported inhibition zones ranging from between 11 to 12 mm using amoxicillin (20 μg) and ampicillin (20 μg) [36,37,38] antimicrobial disks (Appendix A).

The Gram-positive species *S. pyogenes* are one of the main causes of urinary tract infections and poses a significant health concern causing a number of human diseases [39,40]. *S. pyogenes* is also considered to be the most common and significant cause of bacterial tonsillitis in children and adults. Isolates of this bacterial species obtained from the Tikrit Teaching Hospital (Tikrit, Iraq) were all susceptible to the synthesized compounds **22**–**28** at MICs of 12.5 μg/mL, and compounds **27** and **28** did not display any inhibition zones in this test, even at a MIC of 37.5 μg/mL (see Figure 5 and Appendix A).

For the sake of comparison, we compared the antibacterial susceptibility of all the isolates to the five antibacterial agents amoxicillin (20 μg), ampicillin (25 μg), cephalothin (30 μg), azithromycin (15 μg) and doxycycline (30 μg) by the disc diffusion process. Isolates of the three species were susceptible to the five antibiotics and displayed inhibition zones ranging from 5.8–19 mm. *S. pyogenes* isolates showed intermediate zones ranging from 5.8–7.6 mm compared to other Gram-negative species isolates (see Figure 5, Appendix A).

All species included in this research, however, were more susceptible to the synthesized compounds **22**–**28** at higher MICs ranging from 37.5 to 125 μg/mL compared to the control antibiotics. This low activity may, potentially, be caused by resistance-acquiring bacteria due to the arbitrary use of antibiotics by patients [41,42,43].

The aim of this study was also to observe any relation between the structure of the synthesized compounds **22**–**28** and the susceptibility of the bacterial isolates towards them. Thus, as described earlier, the antibacterial susceptibility of compounds **22**–**28** and antibiotics given as MIC values in μg/mL were used to achieve this goal. For compounds **22**–**28**, Figure 3, Figure 4 and Figure 5 clearly indicate that *E. coli* is more susceptible to compounds **22**, **23**, and **12**, while *P. mirabilis* is more susceptible to compounds **22** and **24** as is *S. pyogenes* (see Figure 5). The presence of a *para*-chlorophenyl moiety at the 2-position of the **1** unit of these compounds appears to be responsible for this effect. The presence of a chlorine atom presumably makes such compounds more active than other compounds against the isolates of the three bacterial species. In case of antibiotics, *E. coli* tends to be more susceptible to these five antibiotics, and there were few variations among the isolates’ responses to the antibiotics (see Figure 3, Figure 4 and Figure 5).

As previously mentioned, several pyridine compounds have been shown to possess antibacterial activity [29,30]. In the present study, it appears that the use of aminopyridines in an attempt to enhance the antibacterial activity of the targeted compounds **22**–**28** is not encouraging. Recently, there is a growing interest in the use of drugs of choice as quaternary pyridinium salts to improve their solubility and thus to boost the antibacterial activity due to the presence of quaternary ammonium salts [44].

In order to equate the susceptibility of the more active synthesized compounds, which in this case were compounds **22** and **24** with antibiotics against the bacterial species used in the study, the antibiotics exhibited susceptibility against *E. coli* two-fold greater than compounds **22** and **24**, while antibiotics exhibited susceptibility against *S. pyogenes* two-fold lower than compounds **22** and **24**. Antibiotics demonstrated the same activity against *P. mirabilis* as compounds **22** and **24**. The comparison was made using a concentration of 37.5 µg/mL for compounds **22** and **24** close to the contents of antibiotic discs (see Table 3, Figure 6).

## 3. Materials and Methods

### 3.1. General Information

Melting points were measured using an open capillary on a Buchi melting point apparatus Buchi labortechnik AG, Essen, Germany, and are uncorrected. All the required chemicals used were purchased from Aldrich (Hamburg, Germany). Thin layer chromatography (TLC) was carried out on 5 × 5 plates coated with silica 0.25 cm N-HR/UV_254_ obtained from Merck (Dramstadt, Germany). IR spectra were recorded over a frequency range of 4000–400 cm^−1^ on a FT-IR 8400S spectrophotometer (Shimadzu, Tokyo, Japan). ^1^H-NMR spectra were recorded on a 300 MHz spectrometer (Tehran, Islamic Republic of Iran) using different solvents and TMS as internal reference Chemical shifts are expressed relative to the internal standard on the δ scale in ppm. Elemental analyses (CHN) were determined on an Eur.Vector EA 3000A system (Rome, Italy). Microwave experiments were conducted using a Microwave Synthesis WorkStation (MAS-II, Microwave Chemistry Technology, Shanghai, China)

### 3.2. Chemistry

#### 3.2.1. General procedure for the preparation of derivatives 16–21

Following a previous report [32], an equivalent quantity of 2-chloroacetyl chloride (2 g, 0.0176 mol) was added to a stirred and cooled solution of the appropriate secondary amine pyrrolidine, morpholine, piperidine, 2,4-dimethylaminopyridine, 2-aminopyridine, 3-aminopyridine and 4-aminopyridine (0.0176 mol) in dry dichloromethane (12 mL), containing equivalent quantities of triethylamine (0.0178 mol) as a base. The reaction mixtures were then stirred at room temperature for 1–2 h. Reaction progress was tracked via TLC. Once the reactions were complete, aqueous sodium carbonate was added with shaking until the medium became neutral. The organic layers were washed with water separately and dried over anhydrous magnesium sulfate. The solvents were removed under vacuum and the residual materials treated with ether to give crystalline samples of of 2-chloro (pyrrolidinyl, morpholinyl, piperazinyl, pyperidinyl, pyridin-2-yl, pyridin-3-yl, pyridin-4-yl, 4,6-dimethyl pyridine-2-yl) acetamides **16**–**21**, and recrystallization from different solvents gave pure derivatives **16**–**21**.

2-Chloro-1-(pyrrolidine-1-yl)ethan-1-one (**16**). Grey crystals, m.p., 143–145 °C; (Yield: 2.8 g, 78%); υ_max_ [KBr]: 2986 cm^−1^ (C-H aliphatic); 1635 cm^−1^ (C=O amide); 1452 cm^−1^ (C-N); 707 cm^−1^ (C-Cl).

2-Chloro-1-(piperidin-1-yl)ethan-1-one (**17**). White crystals, m.p., 202–203 °C; (Yield: 2.5 g, 85%); υ_max_ [KBr]: 2945 cm^−1^ (C-H aliphatic); 1708 cm^−1^ (C=O amide); 1506 cm^−1^ (C-N); 746 cm^−1^ (C-Cl).

2-Chloro-1-morpholinoethan-1-one (**18**). Brown crystals, m.p., 198–200 °C; (Yield: 2.7 g, 60%); υ_max_ [KBr]: 2975 cm^−1^ (C-H aliphatic); 1650 cm^−1^ (C=O amide); 1437 cm^−1^ (C-N); 732 cm^−1^ (C-Cl).

2-Chloro-N-(pyridin-3-yl)acetamide (**19**). Brown crystals, m.p., 196–198 °C; (Yield: 4.1 g, 83%); υ_max_ [KBr]: 3500 cm^−1^ (N-H amide); 3040 cm^−1^ (C-H aromatic); 2981 cm^−1^ (C-H aliphatic); 1654 cm^−1^ (C=O amide); 1575 cm^−1^ (C=C); 1477 cm^−1^ (C-N); 740 cm^−1^ (C-Cl).

2-Chloro-N-(pyridin-4-yl)acetamide (**20**). White crystals, m.p., 251–253 °C; (Yield:3.9 g, 80%); υ_max_ [KBr]: 3346 cm^−1^ (N-H amide); 3051 cm^−1^ (C-H aromatic); 2941 cm^−1^ (C-H aliphatic); 1677 cm^−1^ (C=O amide); 1550 cm^−1^ (C=C); 1460 cm^−1^ (C-N); 578 cm^−1^ (C-Cl).

2-Chloro-N-(4,6-dimethylpyridin-2-yl)acetamide (**21**). Brown crystals, m.p., 267–278 °C; (Yield: 2.6 g, 90%); υ_max_ [KBr]: 3307 cm^−1^ (N-H amide); 3074 cm^−1^ (C-H aromatic); 2947 cm^−1^ (C-H aliphatic); 1656 cm^−1^ (C=O amide); 1575 cm^−1^ (C=C); 1545 cm^−1^ (C-N); 743 cm^−1^ (C-Cl).

#### 3.2.2. General Procedure for the Preparation of Derivatives 22–28 

(A) Conventional Method

The compounds **22**–**28** synthesized as previously reported [32] by combining equivalent amounts of the acetamide **16** (0.2 g, 0.0013 mol) with substituted *para*-chloro aniline (0.17 g, 0.0013 mol) in dry acetonitrile (7 mL) containing an equivalent amount of triethylamine as a catalyst. The mixture is heated for 2–3 h at 70 °C, and the progress of reactions tracked by TLC until the starting materials disappeared. Upon reaction completion, the reaction mixture was cooled to room temperature, the solvent was removed under vacuum and the residue taken up in CH_2_Cl_2_ (10 mL), treated with aqueous K_2_CO_3_ until neutralization, the organic layer washed with water, isolated and dried over anhydrous magnesium sulphate. The solvent was removed in vacuo, and the resulting gummy materials treated with ether, resulting in crystalline derivatives of **22** (see Table 1). Compounds **23**–**28** were synthesized similarly by using equivalent amounts of starting materials.

((4-Chlorophenyl)amino)-1-(3-rrolidine-1-yl)ethan-1-one (**22**). Grey crystals, m.p., 183–185 °C; (Yield: 0.17 g, 50%); calculated. C, 60.31; H, 6.28; N, 11.76. C_12_H_15_N_2_OCl Founded. C, 60.01; H, 6.02; N, 11.45; υ_max_ [KBr]: 3132 cm^−1^ (N-H); 3078 cm^−1^ (C-H aromatic); 2989 cm^−1^ (C-H aliphatic); 1679 cm^−1^ (C=O amide); 1583 cm^−1^ (C=C); 1541 cm^−1^ (C-N); 690 cm^−1^ (C-Cl). ^1^H-NMR (DMSO-d_6_): δ 7.20–6.75 (4H, dd, J_2_,_3_ 5.25, J_2,6_ 1.30 Hz, aromatic); 6.32 (1H, s, N-H); 3.36 and 1.60 (8H, 2m, pyrrolidinyl protons); and ppm 3.30 (2H, s, -CH_2_CO-).

1-(Pyrrolidin-1-yl)-2-tosylethan-1-one (**23**). Yellow crystals, m.p., 188–189 °C; (Yield: 0.29 g, 85%); calculated. C, 58.42; H, 6.23; N, 5.24. C_13_H_17_NO_3_S Founded. C, 58.10; H, 6.02; N, 5.11; υ_max_ [KBr]: 3068 cm^−1^ (C-H aromatic); 2931 cm^−1^ (C-H aliphatic); 1676 cm^−1^ (C=O amide); 1562 cm^−1^ (C=C); 1515 cm^−1^ (C-N); 1321 cm^−1^; 1153 cm^−1^; 611 cm^−1^ (S=O); ^1^H-NMR (DMSO-d_6_): δ 7.75–7.42 (4H, dd, J_2_,_3_ 5.35, J_2,6_ 1.30 Hz, aromatic, tolyl protons); 3.20 (2H, s, -CH_2_CO-); 2.90 and 1.45 (8H, 2m, pyrrolidinyl protons), 2.35 ppm (3H, s, CH_3_).

1-(Piperidin-1-yl)-2-tosylethan-1-one (**24**). White crystals, m.p., 204–207 °C; (Yield: 0.27 g, 78%); calculated. C, 61.17; H, 6.73; N, 11.08. C_13_H_17_N_2_OCl Founded. C, 60.85; H, 6.90; N, 10.30; υ_max_ [KBr]: 3298 cm^−1^ (N-H); 3031 cm^−1^ (C-H aromatic); 2977 cm^−1^ (C-H aliphatic); 1647 cm^−1^ (C=O amide); 1515 cm^−1^ (C=C); 1460 cm^−1^ (C-N); 1357 cm^−1^; 1174 cm^−1^; 580 cm^−1^ (S=O); ^1^H-NMR (DMSO-d_6_): δ 7.20–6.70 (4H, dd, J_2_,_3_ 5.30, J_2,6_ 1.45 Hz, aromatic); 6.35 (1H, s, N-H); 3.20 (2H, s, -CH_2_CO-); 2.90 and 1.45 ppm (10H, 2m, piperidinyl protons).

1-Morpholino-2-tosylethan-1-one (**25**). White crystals, m.p., 170–172 °C; (Yield: 0.31 g, 90%); calculated. C, 55.12; H, 6.0; N, 4.94. C_13_H_17_NO_4_S Founded. C, 54.90; H, 6.10; N, 5.08; υ_max_ [KBr]: 3099 cm^−1^ (C-H aromatic); 2974 cm^−1^ (C-H aliphatic); 1679 cm^−1^ (C=O amide); 1581 cm^−1^ (C=C); 1539 cm^−1^ (C-N); 1247 cm^−1^; 1172 cm^−1^; 584 cm^−1^ (S=O); ^1^H-NMR (DMSO-d_6_): δ 7.75–7.42 (4H, dd, J_2_,_3_ 5.25 Hz, J_2,6_ 1.4 Hz, tolyl protons), 4.15 (2H, 2, -CH_2_CO-); 3.70 and 2.50 (8H, mm, morpholinyl protons), 2.40 ppm (3H, s, CH_3_).

*N*-(Pyridin-3-yl)-2-tosylacetamide (**26**). White crystals, m.p., 204–207 °C; (Yield: 0.27 g, 78%); calculated. C, 57.93; H, 4.82; N, 9.65. C_14_H_14_N_2_O_3_S Founded. C, 58.20; H, 5.02; N, 9.50; υ_max_ [KBr]: 3298 cm^−1^ (N-H); 3031 cm^−1^ (C-H aromatic); 2977 cm^−1^ (C-H aliphatic); 1647 cm^−1^ (C=O amide); 1515 cm^−1^ (C=C); 1460 cm^−1^ (C-N); 1357 cm^−1^; 1174 cm^−1^; 580 cm^−1^ (S=O); ^1^H-NMR (DMSO-d_6_): δ 10.61 ppm (N-H, s); δ 10.65 (1H, s, N-H); 9.0 (1H, s, pyridinyl proton); 8.41–7.10 (3H, dd, J_4_,_5_ 5.31, J_5,6_ 5.25 Hz, J_2,6_ 1.5 Hz, pyridinyl proton); 7.70–7.35 (4H, dd, J_2_,_3_ 5.25 Hz, tolyl protons); 4.20 (2H, s, -CH_2_CO-); and 2.35 ppm (3H, s, CH_3_).

*N*-(Pyridin-4-yl)-2-tosylacetamide *(***27**).White crystals, m.p., 251–253 °C; (Yield: 0.26 g, 76%); calculated. C, 57.93; H, 4.82; N, 9.65. C_14_H_14_N_2_O_3_S Founded. C, 57.60; H, 5.0; N, 9.61; υ_max_ [KBr]: 3344 cm^−1^ (N-H); 3080 cm^−1^ (C-H aromatic); 2977 cm^−1^ (C-H aliphatic); 1639 cm^−1^ (C=O amide); 1531 cm^−1^ (C=C); 1461 cm^−1^ (C-N); 1365 cm^−1^; 1103 cm^−1^; 557 cm^−1^ (S=O); ^1^H-NMR (DMSO-d_6_): δ 9.90 (1H, s, N-H); δ 8.41–7.91(4H, dd, J_2_,_3_ 5.31 Hz, J_2,6_ 1.5 Hz, pyridinyl protons); δ 7.65–6.40 (4H, dd, J_2_,_3_ 5.20 Hz, J_2,6_ 1.5 Hz, tolyl protons); δ 4.25 (2H, s, -CH_2_CO-); and 2.32 ppm (3H, s, CH_3_).

*N*-(4,6-Dimethylpyridin-2-yl)-2-tosylacetamide (**28**). Brown crystals, m.p., 212–213 °C; (Yield: 0.21 g, 62%); calculated. C, 60.37; H, 5.66; N, 8.80. C_16_H_18_N_2_O_3_S Founded. C, 60.57; H, 5.42; N, 8.91; υ_max_ [KBr]: 3463 cm^−1^ (N-H); 3085 cm^−1^ (C-H aromatic) ; 2945 cm^−1^ (C-H aliphatic); 1677 cm^−1^ (C=O amide); 1596 cm^−1^ (C=C); 1544 cm^−1^ (C-N); 1336 cm^−1^; 1150 cm^−1^; 524 cm^−1^ (S=O); ^1^H-NMR (DMSO-d_6_): δ 10.40 (1H, s, N-H); 8.30–7.40 (2H, 2s, pyridinyl protons); 7.60–7.45 (4H, dd, J_2_,_3_ 5.25 Hz, J_2,6_ 1.55 Hz, tolyl protons); 4.25 (2H, s, -CH_2_CO-); and 2.45–2.35 ppm (9H, s, 3CH_3_).

(B) Microwave Method

The compounds **22**–**28** synthesized as previously stated [32] by combining the equivalent quantities mentioned in the above experiments, substituted aniline and *para*-toluene sulfonate sodium salt with α-chloroacetamides **16**–**21** in dry acetonitrile (7 mL) containing equivalent quantities of triethylamine as a catalyst. The mixture was heated at 65–70 °C under 400 Watt microwave irradiation for 5–10 min. The reaction progress was monitored with TLC until the starting materials had disappeared. After the reaction was completed, the mixture was cooled to room temperature, the solvent removed under vacuum, and the residue taken in CH_2_Cl_2_ (10 mL) and treated with aqueous K_2_CO_3_ until neutralization, then the organic layer was washed with water, separated and dried over anhydrous magnesium sulfate. The solvent was removed under vacuum, and the resulting gummy products **22**–**28** treated with ether to give crystalline products **22**–**28** (See Table 1).

### 3.3. Antibiotics

Ready-impregnated antibiotic disks of amoxicillin (20 mcg), ampicillin (25 mcg), cephalothin (30 mcg), azithromycin (15 mcg) and doxycycline (30 mcg) were obtained from Samarra Drug Industries, (Samarra, Iraq) and were used as standard broad-spectrum antibacterial agents in the disk diffusion method.

### 3.4. Bacterial Species

Isolates of the three different bacterial species, namely, two Gram-negative bacteria (*E. coli* plus *P. mirabilis*) and one Gram-positive bacterium (*S. pyogenes*) were collected from the Tikrit Teaching Hospital (Tikrit, Iraq). All the isolates of *E. coli* and *P. mirabilis* were collected from urine samples of urinary tract infection (UTI) patients and the isolates of *S. pyogenes* were collected from tonsillitis patients.

### 3.5. Antibacterial Susceptibility Testing

Antibacterial resistance to seven new synthesized compounds **22**–**28** of the acetamide class and five antibiotics, including amoxicillin, ampicillin, cephalothin, azithromycin and doxycycline, was evaluated using the standard disk diffusion method [45] and the diffusion method (well diffusion) for synthesized compounds **22**–**28**. Both studied bacterial species were pre-cultivated on 24 h incubated nutrient agar plates and bacterial suspensions of each pure isolate were prepared for in vitro antibacterial treatment in 0.6 McFarland turbidity nutrient broth tubes. Mueller Hinton agar plates (Oxoid Ltd., Hampshire, UK), 12 cm in diameter, were prepared as directed by the manufacturer and incubated at 37 °C for 24 h. So, a sterile borer was used to make equidistant wells with a diameter of 6 mm. 50 mg of each of the synthetic compounds **22**–**28** was dissolved in DMSO (1 mL), followed by dilution of 2.5, 5, 15, 25, 35, 45 and 50 μL of each to 1 mL DMSO to obtain concentrations of 0.125, 0.250, 0.750, 1.25, 1.75, 2.25 and 2.5 mg/mL which equivalent to 6.25, 12.5, 37.5, 62.5, 87.5, 112.5 and 125 μg/50, respectively. 50 μL of each of the later concentrations were used to assess antibacterial susceptibility and minimum inhibitory concentrations (MICs) by filling the growth media wells with it accompanied by 24-h incubation at 37 °C and growth inhibition monitor in. DMSO did not display any inhibition of bacterial growth.

### 3.6. Analysis of Results

SPSS software (version 20, IBM, Armonk, NY, USA, www.ibm.com/software/analytics/spss) was used to evaluate the effects of the antimicrobial susceptibility study. Mean values and standard deviations for the inhibition zone diameters were determined. The findings were presented as average values ± SD. Statistically standard deviation, variations are calculated relative to normal antibiotics and the amount of dispersion between them.

## 4. Conclusions

2-Chloroacetamides are versatile intermediates in organic synthesis. The reported synthesis method is based primarily on chloroacetylation of amines by chloroacetyl chloride. The simple replacement of a chlorine atom allowed us to prepare many acetamide derivatives through the reaction with amines and *para*-tosyl sodium salt. A series of new acetamide derivatives were successfully synthesized by adding primary and secondary amines to chloroacetyl chloride **15**. With the help of microwave irradiation, we have been able to synthesize seven compounds in an attempt to increase yields and reduce the reaction time. Moderate to good yields and reduction of the reaction time from 2–3 h to a few minutes were achieved. The application of the compounds **22**–**28** against Gram-positive and Gram-negative bacterial species demonstrated encouraging relatively good antibacterial potency in comparison with the used reference antibiotics. Isolates of two of the tested species, namely, *E. coli and P. mirabilis* displaced higher susceptibility than *S. pyogenes*. The results show that among synthetic compounds **22**–**28** compounds **22** and **24** present the most important antibacterial properties bearing *para*-chlorophenyl moiety in the acetamide 2-position of moiety **1**. We have successfully developed a synthetic method has proven to be a fast, environmentally friendly technique with moderate to good performance in microwave irradiation, and high acceleration of reaction rates has been achieved in the presence of Et_3_N as a base.

## Data Availability

The data presented in this study are available in article and Appendix A.

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
