# Peer review of "Microwave-Assisted Synthesis, Structural Characterization and Assessment of the Antibacterial Activity of Some New Aminopyridine, Pyrrolidine, Piperidine and Morpholine Acetamides"

_molecules, 2021, doi:10.3390/molecules26030533_

Round 1

Reviewer 1 Report

The work by A. S. H. Alsamarrai and S. S. Abdulghani is interesting but the manuscript needs a major revision before its publication.

  • The numbering of the compounds in the manuscript should be changed: compounds should be numbered sequentially as appearing in the manuscript.
  • In the Abstract says “The results show that the most important antibacterial properties exercised by the synthetic compounds 9 and 11 bearing para-chlorophenyl moiety incorporated into the 2-position moiety of acetamide 2.” but should say “The results show that the most important antibacterial properties were exercised by the synthetic compounds 9 and 11 both bearing para-chlorophenyl moiety incorporated into the 2-position moiety of acetamide 2.”
  • In the Introduction says “Many processes have reported the use of acetamide chlorides 3-8 as a useful building block for the synthesis of complex heterocyclic compounds.” but should say “Many processes have reported the use of alpha-chloro acetamides 3-8 as useful building blocks for the synthesis of complex heterocyclic compounds.”
  • -In Figure 1 legend says “Structure of acetylamide moeity” but should say “Structure of acetylamide moiety”.
  • In page 1 says “are among the acetamide derivatives of 2 the most widely used drugs in medicines (figure 2) .” but should say “are among the 2 the most widely used drugs in medicine displaying acetylamide moiety (figure 2) .”
  • In page 1 says “Many derivatives of 2 containing secondary cyclic amines linked to 1 or 2- positions that were reported to have become an anticonvulsant.” but should say “Many derivatives of 2 containing secondary cyclic amines linked to 1 or 2- positions were reported to have become an anticonvulsant.”
  • In page 1 says “For instance, anticonvulsant compounds such as 21 and 22 used to treat epilepsy [17], whereas the combination of compounds 23 and 24 used to inhibit kinase enzyme and antihistamines [18].” but should say “For instance, anticonvulsant compounds such as 21 and 22 used to treat epilepsy [17], or the combination of compounds 23 and 24 used to inhibit kinase enzyme and antihistamines [18].”
  • In Figure 2, the chemical structure of atorvastatin is wrong (a carboxylic acid instead of an aldehyde should be displayed) and should be corrected, also lidocaine. Also the name of cpd 23 is wrong (should say Gleevec (Imatinib) instead of Geevec), and the name of cpd 24 (should say ZM447439).
  • In Figure 3, should say Zoplicone instead zoplicon. The name or reported codes for cpds 27 and 28 should be indicated if known.
  • In page 3 says “Synthesis of broad spectrum compounds with antibacterial activity is required because of the ex-tense use of antibiotics in medicine and prevailing resistance in microorganism.” But should say “Synthesis of compounds with broad antibacterial activity is required because of the extensive use of antibiotics in medicine and prevailing resistance in microorganism.”
  • In page 3 says “Nowadays microwave plays an appealing role in organic synthesis. Herein, we report the use of microwave in organic synthesis of seven newly acetamides derivatives 9-15 form reaction of chloroacetyl chloride 1, with primary, secondary amines and para toluene sulphinate sodium salts.” but should say “Nowadays microwave irradiation represents an interesting technique to be applied in organic synthesis. Herein, we report the use of microwave in the synthesis of new seven acetamide derivatives 9-15 by reaction of chloroacetyl chloride 1, with primary, secondary amines and para toluene sulphinate sodium salts.”. Also a reference about the general use of microwave should be added after the first sentence.
  • In Scheme 1: sulphinate salts should be indicated in the reaction conditions b. In the legend should say “Synthetic” but not “Syntheic”.
  • In page 4 says “the acetamide intermediates 3-8 and the new targeted derivatives 9-15 synthesized in this work are depicted in Scheme (1).” But should say “the alpha-chloro acetamide intermediates 3-8 and the new derivatives 9-15 synthesized in this work are depicted in Scheme 1.”.
  • When referring to tables or schemes in the text, they should be indicated in parentheses. For instance in page 4 says “see Table 1” but should say “(see Table 1)”. Also in page 5 says should say “(see Scheme 1 and Table 1)” and “(see Tables 1, 2, and 3 for 1H-NMR data)”. Please correct it accordingly all along the text.
  • In page 4 says “The synthesis of targeted compounds 9-15 was achieved through two routes: conventional heating and microwave irradiation methods.” but should say “The synthesis of targeted compounds 9-15 was achieved through two routes: conventional heating and microwave irradiation.”
  • In Table 3 the entries of “Type of protons” column should be modified so as to be at same height as the corresponding protons of “No. of protons” column.
  • The English language in the following paragraph in page 7 is confusing and should be properly corrected: “ Tests for the gram-negative species, coli and P. mirabilis cause urinary tract infections (UTI) in patients exhibited inhibition zones for compounds 9 and 10 ranging from 6.0 to 8.4 mm at MIC of 12.5 μg/ml. At MIC of 37.5 μg/ml, isolates of these bacterial species were not susceptible to compounds 13 and 15, while inhibiting zones of other compounds ranging from 6–14 mm Table 4. The results for these two gram-negative species exhibited intermediate inhibition zone diameters are in agreement with other authors who reported 11-12 mm using amoxicillin (20 μg) and ampicillin (20 μg) [25, 26, 27] antimicrobial disks Tables 4 and 5.”
  • -In Materials and Methods section, the compounds numbers should be indicated in bold.

Author Response

Response to Reviewer  1

- Point 1: The numbering of the compounds in the manuscript should be changed:    compounds should be numbered sequentially as appearing in the manuscript.

- Response: Please numbering of the compounds have been done all along the text  see line 11

- Pion 2:  In the Abstract says “The results show that the most important antibacterial properties exercised by the synthetic compounds and 11 bearing para-chlorophenyl moiety incorporated into the 2-position moiety of acetamide 2.” but should say “The results show that the most important antibacterial properties were exercised by the synthetic compounds and 11 both bearing para-chlorophenyl moiety incorporated into the 2-position moiety of acetamide 2.”

- Response: line 20

- Point 3: In the Introduction says “Many processes have reported the use of acetamide chlorides 3-8 as a useful building block for the synthesis of complex heterocyclic compounds.” but should say “Many processes have reported the use of alpha-chloro acetamides 3-8 as useful building blocks for the synthesis of complex heterocyclic compounds.”

- Response: line 33

- Point 4: In Figure 1 legend says “Structure of acetylamide moeity” but should say “Structure of acetylamide moiety”.

- Response: line 40

- Point 5: in page 1 says “are among the acetamide derivatives of the most widely used drugs in medicines (figure 2) .” but should say “are among the the most widely used drugs in medicine displaying acetylamide moiety (figure 2) .

- Response : line 42

- Point 6: In page 1 says “Many derivatives of containing secondary cyclic amines linked to 1 or 2- positions that were reported to have become an anticonvulsant.” but should say “Many derivatives of containing secondary cyclic amines linked to 1 or 2- positions were reported to have become an anticonvulsant.”

- Response: Lines 47 and 48

- Point 7: In page 1 says “For instance, anticonvulsant compounds such as 21 and 22 used to treat epilepsy [17], whereas the combination of compounds 23 and 24 used to inhibit kinase enzyme and antihistamines [18].” but should say “For instance, anticonvulsant compounds such as 21 and 22 used to treat epilepsy [17], or the combination of compounds 23 and 24 used to inhibit kinase enzyme and antihistamines [18].

- Response: line 49

- Point 8; In Figure 2, the chemical structure of atorvastatin is wrong (a carboxylic acid instead of an aldehyde should be displayed) and should be corrected, also lidocaine. Also, the name of cpd 23 is wrong (should say Gleevec (Imatinib) instead of Geevec), and the name of cpd 24 (should say ZM447439).

- Response: Please see figure 2

- Point 9: In Figure 3, should say Zoplicone instead zoplicon. The name or reported codes for cpds 27 and 28 should be indicated if known.

- Response: Please see fig.3 compound 11. Sorry names are not indicated.

- Point 10: In page 3 says “Synthesis of broad spectrum compounds with antibacterial activity is required because of the ex-tense use of antibiotics in medicine and prevailing resistance in microorganism.” But should say “Synthesis of compounds with broad antibacterial activity is required because of the extensive use of antibiotics in medicine and prevailing resistance in microorganism.

- Response: lines 60 and 61

- Point 11: In page 3 says “Nowadays microwave plays an appealing role in organic synthesis. Herein, we report the use of microwave in organic synthesis of seven newly acetamides derivatives 9-15 form reaction of chloroacetyl chloride 1, with primary, secondary amines and para toluene sulphinate sodium salts.” but should say “Nowadays microwave irradiation represents an interesting technique to be applied in organic synthesis. Herein, we report the use of microwave in the synthesis of new seven acetamide derivatives 9-15 by reaction of chloroacetyl chloride 1, with primary, secondary amines and para toluene sulphinate sodium salts.”. Also, a reference about the general use of microwave should be added after the first sentence.

- Response: Please line 63-67.

- Point 12: In Scheme 1: sulphinate salts should be indicated in the reaction conditions b. In the legend should say “Synthetic” but not “Synthetic

- Response: line 72

- Point 13; In page 4 says “the acetamide intermediates 3-8 and the new targeted derivatives 9-15 synthesized in this work are depicted in Scheme (1).” But should say “the alpha-chloro acetamide intermediates 3-8 and the new derivatives 9-15 synthesized in this work are depicted in Scheme 1.”.

- Response: line 76

- Point 14: When referring to tables or schemes in the text, they should be indicated in parentheses. For instance in page 4 says “see Table 1” but should say “(see Table 1)”. Also in page 5 says should say “(see Scheme 1 and Table 1)” and “(see Tables 1, 2, and 3 for 1H-NMR data)”. Please correct it accordingly all along the text.

- Response: line 85

- Point 15: In page 4 says “The synthesis of targeted compounds 9-15 was achieved through two routes: conventional heating and microwave irradiation methods.” but should say “The synthesis of targeted compounds 9-15 was achieved through two routes: conventional heating and microwave irradiation.”

- Response: line 86

- Point 16: In Table 3 the entries of “Type of protons” column should be modified so as to be at same height as the corresponding protons of “No. of protons” column.

- Response: Please see table 3

- Point 17: The English language in the following paragraph in page 7 is confusing and should be properly corrected: “ Tests for the gram-negative species, coli and P. mirabilis cause urinary tract infections (UTI) in patients exhibited inhibition zones for compounds and 10 ranging from 6.0 to 8.4 mm at MIC of 12.5 μg/ml. At MIC of 37.5 μg/ml, isolates of these bacterial species were not susceptible to compounds 13 and 15, while inhibiting zones of other compounds ranging from 6 to 14 mm Table 4. The results for these two gram-negative species exhibited intermediate inhibition zone diameters are in agreement with other authors who reported 11-12 mm using amoxicillin (20 μg) and ampicillin (20 μg) [25, 26, 27] antimicrobial disks Tables 4 and 5.”

- Response: Tested compounds 22 and 23 against gram-negative species, E. coli and P. mirabilis cause urinary tract infections (UTI) in patients, exhibited inhibition zones ranging from 6.0 to 8.4 mm at MIC of 12.5 μg/ml, while isolates of these bacterial species were not susceptible to compounds 26 and 28 even at MIC of 37.5 μg/ml. Compounds 24, 25, and 27 showed inhibition zones ranging from 6 to 14 mm at MIC of 37.5 μg/ml (Table 4). These intermediate  results are in agreement with other authors who reported inhibition zones ranging between 11 to12 mm using amoxicillin (20 μg) and ampicillin (20 μg) [36, 37, 38] antimicrobial disks (Tables 4 and 5).

- Point 18: -In Materials and Methods section, the compounds numbers should be indicated in bold.

Response: Please see pages 17 and 18. Compound numbers : 16, 17, 18, 19, 20, 21

Reviewer 2 Report

The authors have improved the manuscript text. However, some corrections are still needed, as follow:

  • Explain the difference of concentrations. In the first version of the manuscript, the authors describe the concentrations of the tested compounds in mg/mL, and now, the concentrations were modified to ug/mL.
  • Please, revise the use of the word "also" in the sentence: “Also, the aim of this study is also to assess the susceptibility…”. Try to replace for: "In addition, ..."
  • Figures 4-5 are confusing. The variable measured on the ordinate axis is missing. The MIC data, measured in ug/ml, are mixed and compared with the zone of inhibition data, measured in mm. Please review these figures.

Author Response

Responses to Reviewer 2

- Point 1: Explain the difference of concentrations. In the first version of the manuscript, the authors describe the concentrations of the tested compounds in mg/mL, and now, the concentrations were modified to ug/mL

- Response: Unfortunately, it was our error in the first submission that we did not note the further dilution of the first concentration in the experimental. Further dilution has been made, as can be seen in the red paragraph. Please go to page 20. 50 mg of each of the synthetic compounds 22-28 was dissolved in DMSO (1 ml), followed by dilution of 2.5, 5, 15, 25, 35, 45 and 50 μL of each to 1ml DMSO to obtain concentrations of 0.125, 0.250, 0.750, 1.25, 1.75, 2.25 and 2.5 mg/ml which equivalent to 6.25, 12.5, 37.5, 62.5, 87.5, 112.5 and 125 μg/50 respectively. 50 μL of each of the later concentrations were used to assess antibacterial susceptibility and minimum inhibitory concentrations (MICs) by filling the growth media wells with it accompanied by 24-hour incubation at 37 oC and growth inhibition monitor in.

- Point 2: Please, revise the use of the word "also" in the sentence: “Also, the aim of this study is also to assess the susceptibility…”. Try to replace for: "In addition, ..."

- Response: line 66-67

-Point 3: Figures 4-5 are confusing. The variable measured on the ordinate axis is missing. The MIC data, measured in ug/ml, are mixed and compared with the zone of  inhibition data, measured in mm. Please review these figures.

- Response: Figures 4-5-6 were corrected, please see the figures.

Reviewer 3 Report

After the review, the paper could be accepted.

Author Response

Thanks a lot for reviewing our work

Best regards

Reviewer 4 Report

              2331/5000       It is clear that the authors wanted to correct the mistakes reported by reviewers. However, in my opinion, the article still needs a series of changes  to be published. Poor English expression persists, making the article difficult to read and understand.
I think the introduction should be completed. Acetamide derivatives with various actions are presented, but there is no emphasis at all on antibacterial derivatives, although the article proposes the evaluation of this type of action.
There is only one (old) bibliographic index (24) that refers to new acetamide derivatives with antibacterial potential, which makes the reasoning of the study quite weak. I think this can be improved. None of the pyridine derivatives given as an example (in order to argue the choice of the structure of the compounds in the article) has antibacterial action. No examples of pyrrolic or morpholine derivatives with antibacterial potential are given. It's hard to understand how the structure of the  compounds was chosen. The authors may consider an abstract graphical. I think it would be interesting and would make it easier to follow how the structure of the compounds proposed for synthesis was chosen.
Also, the need to research new compounds with antibacterial potential is not well argued. The authors claim Synthesis of broad spectrum compounds with antibacterial activity is required because of the ex-tense use of antibiotics in medicine and prevailing resistance in microorganism.which is true, but there are no bibliographic references to support this (can be completed).
In terms of material and method, it is not specified whether the antibiotics used at those single concentrations were ready-impregnated discs (I think they were). The substances were dissolved in DMSO. Has DMSO been tested as a control? It should be specified. Regarding the interpretation of the results, it is interesting that were used 5 standard antibiotics to analyze 7 new compounds, but the fact that the antibiotics were used at a single concentration (is not actually MIC although this term appears in the article for standard antibiotics), and compounds at several concentrations (all differing from those used in reference antibiotics), make it quite difficult to interpret the results.

Figures 7, 8, 9 do not actually show MIC values ​​for the reference antibiotics, they show values ​​of the diameters of the zones of inhibition at a certain concentration for each antibiotic.

Round 2

Reviewer 1 Report

Authors have made most of the suggested changes but some errors still need to be corrected before publication:

  • In Figure 2, the name of cpd. 9 is wrong (should say Gleevec (Imatinib) instead of Geevec(Imatinib)).
  • Celsius degree symbol should be properly indicated all along the text ( ° C)
  • number of cpd. 27 should be in bold in Scheme 1

Author Response

Thanks for the reviewer1 valuable comments.

All Comments had been revised in the manuscript.

This manuscript is a resubmission of an earlier submission. The following is a list of the peer review reports and author responses from that submission.